# Hemophilia A Resulting in Severe Hyperesthesia Due to Extraparenchymal Spinal Cord Hemorrhage in a Young Golden Retriever Puppy

**DOI:** 10.3390/vetsci9110638

**Published:** 2022-11-17

**Authors:** Charlotte Lubbers, Martijn Beukers, Niklas Bergknut, Geert Paes

**Affiliations:** 1Pride Veterinary Centre, Riverside Rd, Derby DE24 8HX, UK; 2IVC Evidensia Dierenziekenhuis Hart van Brabant, Eerste Zeine 112, 5144 AM Waalwijk, The Netherlands

**Keywords:** hemophilia A, hematorrhachis, hyperesthesia

## Abstract

**Simple Summary:**

This case report describes a case of a male Golden Retriever puppy with spinal cord bleeding caused by hemophilia A. Hemophilia A is a congenital clotting disorder in which there is a lack of one of the clotting factors (factor VIII). It can cause spontaneous bleeding in the skin, muscles, and joints, and, in rare circumstances, also in the vertebral canal. Spinal cord bleeding most frequently results in neurologic deficits such as weakness or even paralysis of one or more limbs, but it has also been associated with spinal pain. Our case presented with severe, generalized pain and mild hindlimb weakness. Computed tomography findings of the spinal cord showed changes compatible with hematomas in the vertebral canal. A diagnosis of hemophilia A was made based on abnormal clotting test results and a significantly decreased factor VIII activity (FVIII:C). Currently, there is no treatment for hemophilia A, so medical management is mainly supportive, and aimed at stopping severe bleeding episodes. Even though the dog responded well to analgesic treatment, the prognosis for severely affected dogs is guarded as recurrent bleeding is often seen. Because of this, the owner of the puppy elected humane euthanasia.

**Abstract:**

A ten-week-old male Golden retriever puppy was presented with severe hyperesthesia, mild neurological deficits and episcleral bleeding. Clotting times showed a normal prothrombin time (PT) and prolonged activated partial thromboplastin time (aPPT). Computed tomography (CT) of the vertebral column showed intradural, extraparenchymal hyperattenuating changes on precontrast CT images and epidural mass lesions, suggestive of hematorrhachis. Hemophilia A was confirmed by a low-factor VIII activity (FVIII:C). Although the dog improved clinically with intravenous analgesia and cage rest, it was euthanized by the owners’ choice because of the risk of developing future episodes of spontaneous hemorrhage. In young male puppies with severe hyperesthesia and mild neurological deficits, hemophilia A should be considered as a possible differential diagnosis.

## 1. Introduction

Hemophilia A is an inheritable coagulation disorder in both humans and dogs causing an increased risk of bleeding. It is caused by a quantitative genetic deficiency in clotting factor VIII (FVIII). The gene coding for factor VIII is located on the X chromosome and because it inherits recessively, mostly male individuals are affected while females are asymptomatic carriers [1]. However, de novo mutations can occur, leading to an acquired form of the disorder. Furthermore, factor VIII deficiency can also be acquired by Angiostrongylus vasorum infection [2]. Hemophilia A has been identified in many different purebred dogs, as well as in unrelated mixed breeds [1,3].

Clinical signs of hemophilia A are caused by excessive or uncontrolled bleeding, which can be triggered by minor events or even occur spontaneously. Most common findings are typical for a secondary hemostatic disorder and include subcutaneous or intramuscular hematomas, prolonged bleeding after surgery, prolonged bleeding associated with loss of deciduous teeth, abnormal bleeding from minor wounds or injection sites, bleeding in the body cavity or organs and lameness caused by hemarthrosis [3,4,5]. Hemophilia A resulting in hematorrhachis and neurological deficits in young dogs has been reported in the past, with clinical signs mainly of a neurological nature, such as acute paraplegia or tetraplegia, combined with areflexia, incontinence and lack of superficial or even deep pain perception [6,7,8]. Spinal cord hemorrhage secondary to hemophilia A resulting in pain or hyperesthesia has infrequently been reported. One older study describes a one-year-old Labrador Retriever–Terrier mix with cervical pain and paraparesis, and a recent publication reports three young, male dogs with paraspinal hyperesthesia, caused by vertebral canal hemorrhage resulting from hemophilia A [8,9]. Two of the dogs in the latter study presented with cervical hyperesthesia, which was not associated with neurologic deficits in one dog, while the other dog exhibited progressive neurologic deficits, eventually resulting in severe non-ambulatory tetraparesis. The dog without neurologic deficits developed hyperacute extradural hemorrhage post cerebrospinal fluid (CSF) sampling, resulting in multiple cranial nerve deficits with a need for mechanical ventilation. Hematorrhachis in all three cases was diagnosed by magnetic resonance imaging (MRI).

A suspicion of hemophilia A typically arises when coagulation tests show a prolonged activated partial thromboplastin time (aPPT) or activated clotting time (ACT) with a normal prothrombin time (PT), thrombin time (TT), and fibrinogen. A definitive diagnosis is made by measurement of the coagulation factor VIII activity (FVIII:C), which will be decreased. Spinal cord hemorrhage in dogs can be diagnosed either by myelography, computed tomography (CT), or MRI [8,9,10,11,12,13,14]. Other reported causes for spontaneous hemorrhage in the vertebral canal in dogs are trauma, intervertebral disk disease, vascular anomalies, Angiostrongylus vasorum infections, steroid-responsive meningitis-arteritis (SRMA), and other coagulopathies [7,10,11,12,13,14,15,16]. Based on the classification in human medicine, hemophilia in dogs is classified as mild (FVIII:C ≈ 6–20%), moderate (FVIII:C ≈ 2–5%), or severe (FVIII:C < 2%) [3,17]. In humans, the degree of severity of clinical signs is related to the degree of decrease in FVIII:C. Although it has been suggested that this is also true in dogs [1], more recent publications show that this does not seem to be the case and that, therefore, the FVIII:C cannot be used as a prognostic indicator in dogs with hemophilia A [3,18].

Treatment of hemophilia A depends on the severity of hemorrhage episodes. Transfusions may be needed in case of too much blood loss or to replace the deficient factor in order to stop bleeding. Either fresh frozen plasma (FFP) or cryoprecipitate must be used because they contain FVIII [19]. Both FFP and cryoprecipitate are equally effective in increasing FVIII:C, but FFP is associated with more adverse reactions and has a higher risk of causing volume overload [20]. Limitations of factor-based therapy include the short half-life of FVIII [21] and the formation of inhibitory alloantibodies [22]. In the last decades, gene therapy has gained growing interest and dogs have been used as animal models for humans, because of the similar expression of hemophilia A in both species. Two recent studies showed successful treatment of, respectively two and nine hemophilia A dogs with adeno-associated viral (AAV) gene therapy resulting in the correction of the FVIII deficiency to 1%−11.3% of normal FVIII levels [21,23].

## 2. Case Description

A ten-week-old male Golden retriever was referred to the neurology department of referral hospital ‘Hart van Brabant’ in Waalwijk (The Netherlands), with severe hyperesthesia and difficulty walking. The complaints started the day before, when the dog had been locked in a kennel for one hour while the owner was out. Radiographs of the spinal cord and the pelvis, made by the referring veterinarian, did not show any radiographic abnormalities. Treatment was started with meloxicam (Boehringer Ingelheim, Alkmaar, The Netherlands) 0.2 mg/kg via subcutaneous injection (continued at 0.1 mg/kg via oral administration) which did not result in improvement of the clinical signs, so the dog was referred for further examination.

The dog was presented to us with a kyphosis, low-head carriage, palmigrade and plantigrade stance, and could barely walk or be touched due to extreme pain. A physical examination showed no abnormalities, except for a small unilateral episcleral hemorrhage in the right eye and severe pain on palpation of the vertebral column, chest wall, flanks, and extremities, manifested by vocalizing and biting. Systolic blood pressure was 130 mmHg. Neurological examination showed mild ataxia, reduced withdrawal reflexes in the pelvic limbs and weak patellar reflexes. The rest of neurological reflexes and responses were normal. Orthopedic examination did not reveal joint effusion or immobility, but a comprehensive exam could not be completed due to the severe pain.

The dog was hospitalized and started on an intravenous constant rate infusion (CRI) of ketamin (AST Farma, Oudewater, The Netherlands) 1 mg/kg/h and a CRI of fentanyl (Dechra, Bladel, The Netherlands) 2 µg/kg/h. A complete blood examination was performed, including a complete blood count (CBC), biochemistry, electrolytes, CK (creatinine kinase), and CRP (C-reactive protein). There were no significant abnormalities found. Clotting times showed a normal prothrombin time (PT) and prolonged activated partial thromboplastin time (aPPT) (142.0 s; RI: 72.0 to 102.0 s). D-dimers were slightly elevated (0.7 µg/mL; RI: 0.023 to 0.65 µg/mL) (Table 1).

A CT-scan of the vertebral column showed diffuse intradural, extra-parenchymal hyperattenuation on precontrast images (Figure 1) and increased meningeal contrast enhancement (Figure 2a,b). In addition, there were multiple epidural mass lesions in the thoracic and lumbar spine (T6–T10; L2–L5), causing mild extradural cord compression (Figure 3). There was no evidence of local lymphadenopathy. Given the presence of scleral hemorrhage, the prolonged aPPT, and the hyperattenuating nature of the changes, multifocal intradural/subdural and epidural hemorrhage, secondary to a coagulopathy, was considered most likely [24]. In order to find the underlying cause of the coagulopathy, coagulation factor activity VIII (FVIII:C) and IX (FIX:C) were evaluated (Table 1) and the dogs were tested for Angiostrongylus vasorum. FVIII:C was significantly decreased (6%, RI: 60–150%), confirming hemophilia A. At initial presentation, FIX:C was also decreased (35; RI: 60–150%), however, repeat testing five days later showed a normal FIX:C with a persistently decreased FVIII:C. A faecal Baerman was negative for Angiostrongylus Vasorum.

The dog improved clinically with strict cage rest and analgesic medication including ketamin and fentanyl CRI, as described above, and gabapentin (Pharmachemie BV, Haarlem, The Netherlands) 10 mg/kg every eight hours, and prednisolone (AST Farma, Oudewater, The Netherlands) 1 mg/kg every 24 h. Follow-up blood work showed stable hematocrit and protein levels. This, together with the fact that the clinical signs were improving, suggested that ongoing active bleeding was unlikely. Because of this and because the owner expressed financial concerns, it was decided to not administer blood products to the dog. Even though clinical improvement was observed, it was discussed with the owner that the dog was at high risk for developing future spontaneous bleeding episodes. In the end, because of the guarded long-term prognosis, the owner elected humane euthanasia.

## 3. Discussion

Signs of lameness and pain are commonly described with hemophilia A, but they are usually caused by hemarthrosis [3,4,5]. The absence of joint effusion made hemarthrosis highly unlikely in this dog. While severe pain was the primary complaint in the reported case, the dog also showed hind limb paresis, reduced reflexes and ataxia, raising suspicion for the presence of hematorrhachis.

Hemophilia A resulting in hematorrhachis and neurological deficits in young dogs has been reported in various case reports. In most of these reports, clinical signs were predominantly of neurological origin [6,7]. In the reported case, the dog presented with hyperesthesia as the main complaint, and only mild neurological deficits. Recently, two similar cases have been described by Fowler et al. [9] in which both dogs presented with paraspinal hyperesthesia in the first instance. In one of these cases, additional neurological deficits (more pronounced than in our case) developed over the course of one month, while in the other case hyperacute hemorrhage caused rapid deterioration within a few hours, leading to cranial nerve deficits and the need for mechanical ventilation. In our case, neurological signs had an acute onset, were very mild, and did not progress in the next week. However, because this dog was euthanized, we cannot say if these deficits would have progressed over time. In the previously reported cases, the pain described was limited to manipulation or palpation of the spinal area. In our case, the pain was more generalized as it was elicited on not only the palpation of the thoracic and lumbar vertebrae, but also on palpation of the flanks, abdominal wall, and extremities.

Possible causes for hyperesthesia in combination with hindlimb paraparesis are thoracolumbar intervertebral disc disease, degenerative myelopathy, discospondylitis, meningomyelitis, spinal cord neoplasia, or spinal trauma [25]. Degenerative disease or neoplasia was highly unlikely considering the young age of the dog. Radiographs of the vertebral column did not show any fractures, nor were there signs indicative of discospondylitis. Although radiographic changes associated with discospondylitis can appear two weeks or more after the onset of clinical signs [26,27], non-specific changes such as unsharp, irregular cortical margins of the vertebral bodies, may be seen at an earlier stage [26,28]. Herniated disk material may also appear on CT images as a mildly hyperattenuating mass effect [29]. In our case there were multiple masses making a hemorrhage or an infection, causing more obvious differentials. Because of the presence of episcleral bleeding and prolonged aPPT, it was decided not to obtain a cerebrospinal fluid sample because of the higher risk of complications in case of a coagulopathy. CSF sampling is not without risk in dogs with a coagulopathy, which was also demonstrated in one of the cases from the study of Fowler et al. [9], in which CSF sampling resulted in hyperacute extradural hemorrhage resulting in multiple cranial nerve deficits and need for mechanical ventilation. Because the owners declined a post-mortem exam, the additional presence of a meningomyelitis cannot be completely ruled out. However, the peracute onset of signs, the absence of fever, the absence of an inflammatory leukogram, and the fact that C-reactive protein (CRP) was not elevated, made an infectious or inflammatory cause unlikely in this case [30]. An epidural hemorrhage has been described as secondary to steroid-responsive-arteritis-meningitis [11,13,14], but in these cases, abnormal coagulation times were not present.

Hyperesthesia can be a result of inflammatory disease affecting the meninges, nerve roots, bone, and periosteum, which are innervated by nociceptive fibers [8]. Therefore, the intradural/subdural and extra-parenchymal hemorrhage could explain the localized pain complaints in this case. However, it does not explain the severe pain response on palpation of the flanks, abdominal wall, and extremities. In human medicine, it is described that sensitization of a peripheral nerve can occur after inflammation, leading to a reduced threshold for the activation and hyperexcitability of primary afferent neurons [31]. This can contribute to pain hypersensitivity within the innervation zone of an affected nerve. The spinal cord compressions in this case were present in the regions T6–T10 and L2–L5, of which the dermatomes cover parts of the abdominal wall and flanks. We hypothesize that the severe pain reaction of touching the extremities could be a result of anticipation of pain when moving them. The palmigrade and plantigrade stance of the dog could suggest a myopathy or carpal laxity syndrome. However, it would not be expected that these conditions would improve with only analgesic treatment.

Possible causes for episcleral bleeding are trauma or a coagulopathy. Episcleral bleeding is more typical in primary hemostasis disorders but can be seen in secondary hemostasis abnormalities as well [32]. The normal platelet count with an increased aPPT, confirmed a secondary hemostasis disorder. Differentials for an increased aPPT with a normal PT are factor deficiencies in factor VIII (hemophilia A), XI (hemophilia B), XI (hemophilia C), and XII (Hageman trait) [19]. Because, to the author’s knowledge, Hageman trait has only been described in cats and hemophilia C is very rare in dogs, it was decided to only measure factor activity VIII (FVIII:C) and IX (FIX:C), which initially were both decreased. After five days, factor assays were repeated and FVIII:C was still decreased while FIX:C had increased. This confirmed the diagnosis of hemophilia A. The initial decrease in FIX:C was likely caused due to the consumption of this factor secondary to hemorrhage. The increased D-dimers were also thought to be secondary to hemorrhage [33].

Hemophilia A has been described in a Golden retriever family before [34]. The dogs affected did, however, not show signs of hematorrhachis. Reported signs included subcutaneous hematoma, prolonged bleeding from deciduous teeth eruption sites, prolonged bleeding after injection or surgery, abnormal bleeding from minor wounds, and lameness due to hemarthrosis. Currently, genetic testing for hemophilia A is not available for this breed.

In humans, MRI is considered the gold-standard investigation for spinal cord hematomas [35]. Most recent studies in veterinary medicine also use MRIs to differentiate a spinal cord hemorrhage from other neurological disorders, but CTs and myelography can be used for this purpose as well [7,8,9,10,11,12,13,14]. We opted for a CT-scan in this case, mainly for practicality, being available out of hours, as well as financial reasons. Although pet insurance has become more common worldwide in the last decades, financial constraint is still one of the reasons to decline diagnostics and/or treatment in veterinary medicine [36]. CT-findings in the case of hemorrhage can be increased attenuation in the subarachnoid or subdural space, and contrast enhancement or spinal cord and/or meninges [8,11]. Recognizing these in combination with findings compatible with a clotting disorder make a CT-scan a valuable diagnostic tool in dogs that are suspected of spinal cord hemorrhage caused by a coagulopathy, in practices without access to MRI or for owners that are financially constrained.

Treatment for hemophilia A, currently, is largely supportive, with the goal to stop severe bleeding episodes. In this dog it was decided to not administer blood products, partially because of financial reasons, but also because the dog improved clinically with analgesic therapy and because the hematocrit and protein levels remained stable, suggesting that active, ongoing bleeding was unlikely. It is possible that if this dog would have received a transfusion with FFP or cyroprecipitate at the start of treatment, the clinical signs might have subsided quicker. Two of the three recently described dogs with pain secondary to spinal cord hemorrhage due to hemophilia A, were treated with blood products. One of these dogs received a packed red blood cell transfusion as well as cryoprecipitate, while the other received an FFP transfusion. In one of these dogs, oral analgesia was sufficient to improve the pain complaints, while the other dog required decompressive surgery with removal of a compressive extradural hematoma [9]. In humans, a common complication, after giving FVIII to a patient with hemophilia A, is the formation of inhibitory alloantibodies. In dogs, this has rarely been reported. an FVIII inhibitor will complicate the treatment of bleeding episodes in hemophilia A, leading to a larger volume of blood products needed to stop it [22].

In recent years, adeno-associated virus (AAV)-based liver-directed gene therapy has shown promising results in both veterinary and human medicine, with two studies showing the sustained expression of increased FVIII levels and decreased bleeding episodes in hemophilia A dogs treated with gene therapy [21,23]. Possible side effects of gene therapy include immunological response, liver toxicity, failure of therapy, and the risk of developing hepatocellular carcinoma [37]. However, long-term follow-ups, over a period of ten years, of nine hemophilia A dogs treated with AAV gene therapy showed that none of the dogs had evidence of tumor growth or altered liver function [23].

The prognosis for dogs with hemophilia A varies and is, therefore, difficult to predict. One study evaluated the clinical outcome of a group of dogs with hemophilia A in which the median follow-up time was 11 months (range, one month to ten years), suggesting an acceptable long-term outcome in some cases [3]. Reviewing the literature, survival times varying from five to 14 months have been reported [8,9,18]. It is safe to say that some cases with hemophilia A can be maintained; however, careful counseling of the owner is advised because managing a dog with hemophilia A requires more need for veterinary care leading to a higher financial burden. This, and because the owners did not want their dog to suffer again from extreme pain due to possible future bleeding, lead, in our case, to the decision of humane euthanasia.

## 4. Conclusions

To the authors' knowledge, this is the second case report describing the CT-imaging of a spinal cord hemorrhage caused by a congenital coagulopathy, and for the first time, this is resulting in severe, generalized hyperesthesia as the primary clinical sign without progressively worsening neurological deficits. Hemophilia A should, thus, be considered in the differential diagnosis of extreme pain and mild neurological dysfunction associated with intradural, extra-parenchymal hyperattenuating changes on pre-contrast CT images and epidural mass lesions in young, especially male, dogs. In these cases, especially when concurrent bleeding in other locations is observed, coagulation testing is advised. When coagulation tests show a prolonged activated partial thromboplastin time (aPPT) or activated clotting time (ACT) with a normal prothrombin time (PT), thrombin time (TT), and fibrinogen, hemophilia should be suspected. Subsequently, measurement of the coagulation factor VIII activity (FVIII:C) for hemophilia A and factor IX activity (FIX:C) for hemophilia B is recommended.

## Figures and Tables

**Figure 1 vetsci-09-00638-f001:**
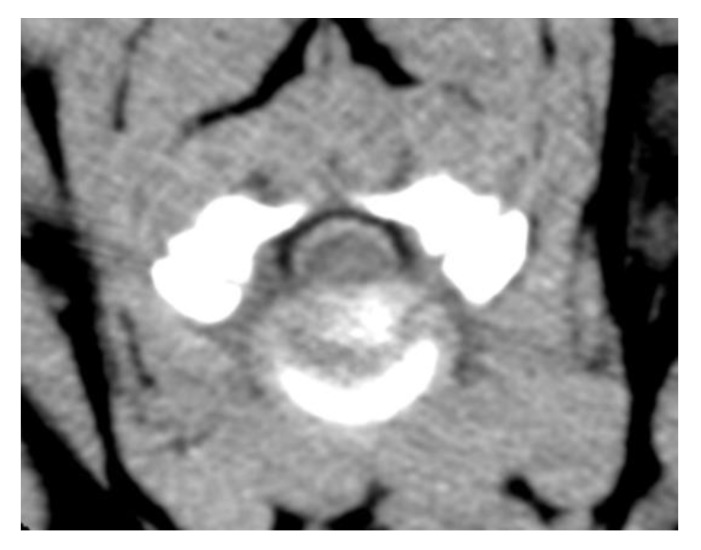
Transverse precontrast CT image at the level of C3–C4, showing intradural, extraparenchymal hyperattenuating changes. Reconstructed with soft tissue algorithm, window width (WW) 150, window level (WL) 35.

**Figure 2 vetsci-09-00638-f002:**
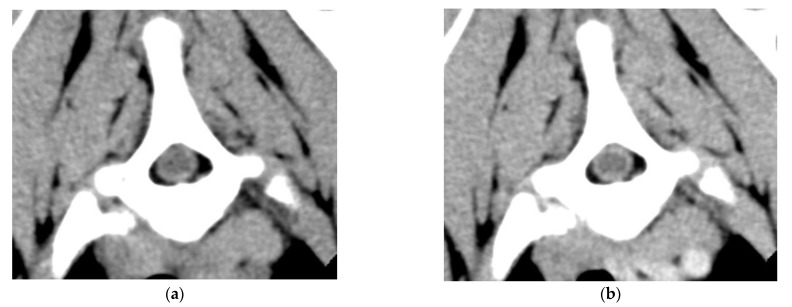
Transverse images at the level of Th2: (**a**) Pre-contrast (**b**) Post-contrast. Reconstructed with soft tissue algorithm, demonstrating an increased meningeal contrast enhancement (WW 150, WL 35).

**Figure 3 vetsci-09-00638-f003:**
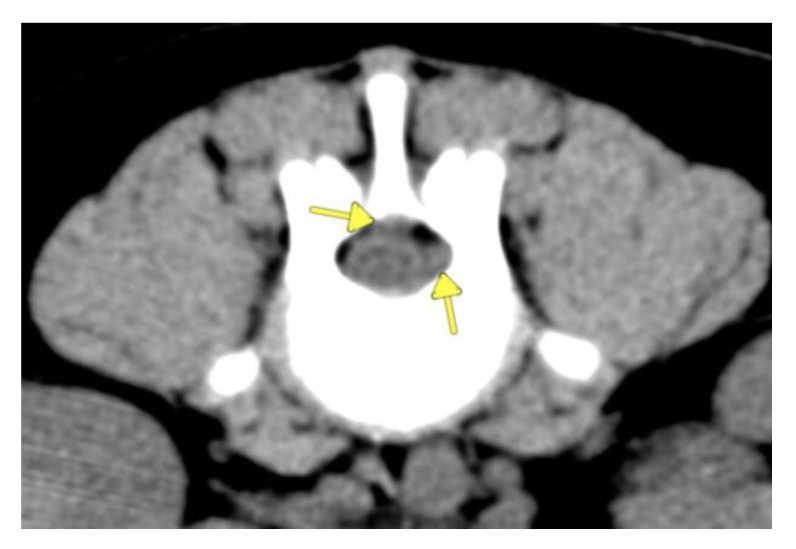
Transverse pre-contrast image at the level of L3, showing a small extradural mass lesion (arrows). Reconstructed with soft tissue algorithm, WW 150, WL 35.

**Table 1 vetsci-09-00638-t001:** Coagulation assay results.

Measurement	1	2	Reference Range
PT (s)	13.0	11.0	11.0–17.0
aPPT (s)	142.0	187.0	72.0–102.0
FVIII:C (%)	6	7	60–150
FXI:C (%)	35	60	60–150
D-dimers (µg/mL)	0.7	-	0.023–0.65

PT = prothrombin time; aPPT = activated partial thromboplastin time; FVIII:C = factor VIII activity; FIX:C = factor XI activity.

## Data Availability

Not applicable.

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
