# Peer review of "Hemophilia A Resulting in Severe Hyperesthesia Due to Extraparenchymal Spinal Cord Hemorrhage in a Young Golden Retriever Puppy"

_vetsci, 2022, doi:10.3390/vetsci9110638_

Round 1
Reviewer 1 Report
The manuscript analyzed the causes of severe hyperesthesia in young golden retrievers with hemophilia A through coagulation assay and CT-scan of the vertebral column. I suggest a revision.
1. What is the clinical significance of the study? Please discuss more questions to increase the meaning of the article.
2. The authors concluded that “To the authors knowledge, this is the first time haemophilia A causing spinal cord hemorrhage is described in a young Golden Retriever with severe hyperesthesia being the primary clinical sign.” Recent studies have revealed the relationship between hemophilia A and hyperesthesia. For example, PMID: 35498741. Authors can discuss more on the relationships on puppy.
3. Beside, did the authors perform more testing after euthanasia? For example, additional Lumbar cerebral spinal fluid sampling can help identify meningitis.
4. CRI should have its full name when it is first mentioned in this manuscript.
5. The key factor for confirming hemophilia A needs to be briefly discussed in the Results section, so that the logic can be a better stated.
6. The References need to be significantly improved. The manuscript cited many old references.
7. The manuscript used two names hemophilia and haemophilia when writing about hemophilia A, and the format should be unified.
Reviewer 2 Report
Lubbers et al. report on a case of hyperesthesia in an young male Golden Retriever puppy. Imaging diagnosis suggested haematorrhachis. Deficiency of factor VIII activity confirmed Hemophilia A as a cause of coagulopathy. This is the second manuscript reporting Haemophilia A as an underlying cause of hyperesthesia. Scientific merit is low compared to the first report, however it is of interest to the veterinary community to obtain additional information regarding this rare coagulopathy.
Line 122: Please cite the following report:
- 2022 Apr 15;9:871029.
doi: 10.3389/fvets.2022.871029. eCollection 2022.
Clinical, Diagnostic, and Imaging Findings in Three Juvenile Dogs With Paraspinal Hyperesthesia or Myelopathy as a Consequence of Hemophilia A: A Case Report
Affiliations- PMID: 35498741
- PMCID: PMC9051508
- DOI: 10.3389/fvets.2022.87102
Line 120: "humane" instead of "human"
Line 157: "localized" instead of "lokalized"
Line 91: please comment on the choice of CT scan instead of MRI.
Round 2
Reviewer 1 Report
The manuscript has been improved a lot. However, the clinical significance can be improved and the manuscript still needs revision.
1. Since the authors only did some primary experiments or examination, I still suggest they should at least discuss more about mechanisms of this case. What is the current treatment method for hemophilia A in puppy? Are there any successful treatment cases?
2. In the paragraph of “Treatment for hemophilia A …”, liver-targeting AAV-mediated gene therapy must be introduced. References such as PMID: 33199875, PMID: 34232980, PMID: 32258217, should be cited.
3. In conclusion, the authors could also give more advice, for example, future measurement of coagulation factors.
